# Critical Metal Particles in Copper Sulfides from the Supergiant Río Blanco Porphyry Cu–Mo Deposit, Chile

Jorge Crespo [1,2,*] , Martin Reich [1,2] , Fernando Barra [1,2], Juan José Verdugo [3] and Claudio Martínez [3]

1   Department of Geology and Andean Geothermal Center of Excellence (CEGA), Facultad de Ciencias Físicas y Matemáticas, Universidad de Chile, Plaza Ercilla 803, Santiago 8370450, Chile; mreich@cec.uchile.cl (M.R.); fbarrapantoja@ing.uchile.cl (F.B.)

2   Millennium Nucleus for Metal Tracing Along Subduction, Facultad de Ciencias Físicas y Matemáticas, Universidad de Chile, Santiago 8370450, Chile

3   Codelco División Andina, Avenida Santa Teresa, N° 513 Los Andes, V Región 2102660, Chile; jverdugo@codelco.cl (J.J.V.); cmart006@codelco.cl (C.M.)

*   Correspondence: jorge.crespo@ug.uchile.cl; Tel.: +56-9-8978-3139

**Abstract:** Porphyry copper–molybdenum deposits (PCDs) are the world's most important source of copper, molybdenum and rhenium. Previous studies have reported that some PCDs can have sub-economic to economic grades of critical metals, i.e., those elements that are both essential for modern societies and subject to the risk of supply restriction (e.g., platinum group elements (PGE), rare earth elements (REE), In, Co, Te, Ge, Ga, among others). Even though some studies have reported measured concentrations of Pd and Pt in PCDs, their occurrence and mineralogical forms remain poorly constrained. Furthermore, these reconnaissance studies have focused predominantly on porphyry Cu–Au deposits, but very limited information is available for porphyry Cu–Mo systems. In this contribution, we report the occurrence of critical metal (Pd, Pt, Au, Ag, and Te) inclusions in copper sulfides from one of the largest PCDs in the world, the supergiant Río Blanco-Los Bronces deposit in central Chile. Field emission scanning electron microscope (FESEM) observations of chalcopyrite and bornite from the potassic alteration zone reveal the presence of micro- to nano-sized particles (<1–10 μm) containing noble metals, most notably Pd, Au, and Ag. The mineralogical data show that these inclusions are mostly tellurides, such as merenskyite ((Pd, Pt) (Bi, Te)$_2$), Pd-rich hessite (Ag$_2$Te), sylvanite ((Ag,Au)Te$_2$) and petzite (Ag$_3$AuTe$_2$). The data point to Pd (and probably Pt) partitioning in copper sulfides during the high-temperature potassic alteration stage, opening new avenues of research aimed at investigating not only the mobility of PGE during mineralization and partitioning into sulfides, but also at exploring the occurrence of critical metals in porphyry Cu–Mo deposits.

**Keywords:** platinum-group elements; silver; gold; Pd-tellurides; porphyry Cu–Mo; Rio Blanco-Los Bronces; Chile

## 1. Introduction

Porphyry Cu–Mo deposits (PCDs) are typically associated with calc–alkaline intrusive rocks, and currently provide 60% of world's copper supply [1]. In addition to Cu, PCDs are major sources of Mo and Re [2], and previous studies have reported the presence of relevant amounts of other metals including Au, Ag, Se, Te, U, W, Bi, Co, and platinum group elements (PGE) [3–11]. Some of these elements are considered as "critical" for the expected growth of renewable energy technology, and thus,

they are both essential for modern societies and subject to the risk of supply restriction [12–14]. Despite their strategic importance, surprisingly few trace element concentration data are available for sulfides in PCDs when compared to other deposits such as orogenic, epithermal, Carlin-type Au deposits, and volcanogenic massive sulfide (VMS) deposits [15–18]. Only a small number of studies have constrained the concentrations of trace metals in sulfides from PCDs using modern micro-analytical techniques (e.g., [7,19–23]); hence, many questions still remain concerning the speciation and mineralogical form of trace metals within sulfides.

Among trace metals, gold and silver are relevant byproducts in porphyry Cu–Mo deposits, with estimated median grades of 9 ppb Au and ~1 ppm Ag [24]. Previous studies have shown that Au is preferentially incorporated into bornite and chalcopyrite within potassic alteration zones, with concentrations reaching ~2–4 ppm in chalcopyrite and ~80–364 ppm in bornite [7,25]. Silver, on the other hand, has been reported to reach up to hundreds or thousands of ppm in bornite, chalcopyrite, and chalcocite in porphyry-type deposits in northern Chile [26,27]. A recent study in PCDs from Romania shows that Ag in bornite is an order of magnitude greater than in chalcopyrite, and two orders of magnitude higher than in pyrite [20].

Anomalous PGE concentrations have been documented in a number of PCDs worldwide. Tarkian et al., 1999 [4] reported PGE data from sulfide, and flotation concentrates from 33 PCDs from Chile, Peru, Argentina, US, Canada, Indonesia, and Papua New Guinea. The cited study reveals that PCDs can contain relatively high Pd concentrations (130–1900 ppb) that are correlated with high Au contents (1–28 ppm). Tarkian et al., 1999 [4] also identified the presence of platinum group minerals (PGM), which occurred as micrometer-scale inclusions in chalcopyrite. Pašava et al., 2010 [9] reported an average Pd value of 55.2 ppb in flotation concentrates from the Kalmakyr porphyry Cu–Au–Mo deposit in Uzbekistan and where the concentration of Pd and Au show a strong correspondence with copper. Palladium was also found associated with Ag, Se, and S in two alkaline porphyry Cu–Au deposits (the Afton and Mount Milligan deposits) in the Canadian Cordillera [28]. The cited authors documented that at least 90% of the bulk Pd + Pt occurs within pyrite, and it is not related to copper sulfides. Both Pd and Pt are highly enriched in the cores of the pyrite grains (up to 90 ppm and 20 ppm, respectively) and their abundance correlates well with the Co content in pyrite (up to 4 wt %). These large concentrations of trace metals have not been observed directly in porphyry Cu–Mo deposits, an attribute that has led to overlooking their PGE potential.

Here, we document that the occurrence of noble metal (Au, Ag, and Pd)-bearing mineral inclusions in copper sulfides within the potassic alteration zone of the world-class Río Blanco porphyry Cu–Mo deposit in central Chile. We combined observations using field emission scanning electron microscopy (FESEM) and electron microprobe analyses (EMPA) of selected grains of chalcopyrite and bornite. Our results show that PGE minerals may be more frequent in PCDs than previously thought, and that high-resolution FESEM techniques greatly facilitate the imaging and characterization of micrometer to nanometer-sized particles in copper sulfides.

## 2. Geology of the Río Blanco Deposit

The Río Blanco PCD is located ~60 km northeast of Santiago, at an altitude of between 3700 and 4300 m above sea level in the high Andes of Central Chile (Figure 1A). From north to south, mining activities are focused on three areas, i.e., the Río Blanco (RB) underground mine, and the Don Luis (DL) and Sur–Sur (SS) open pits (Figure 1B). In 2017, the mine produced 220,000 tons of fine copper.

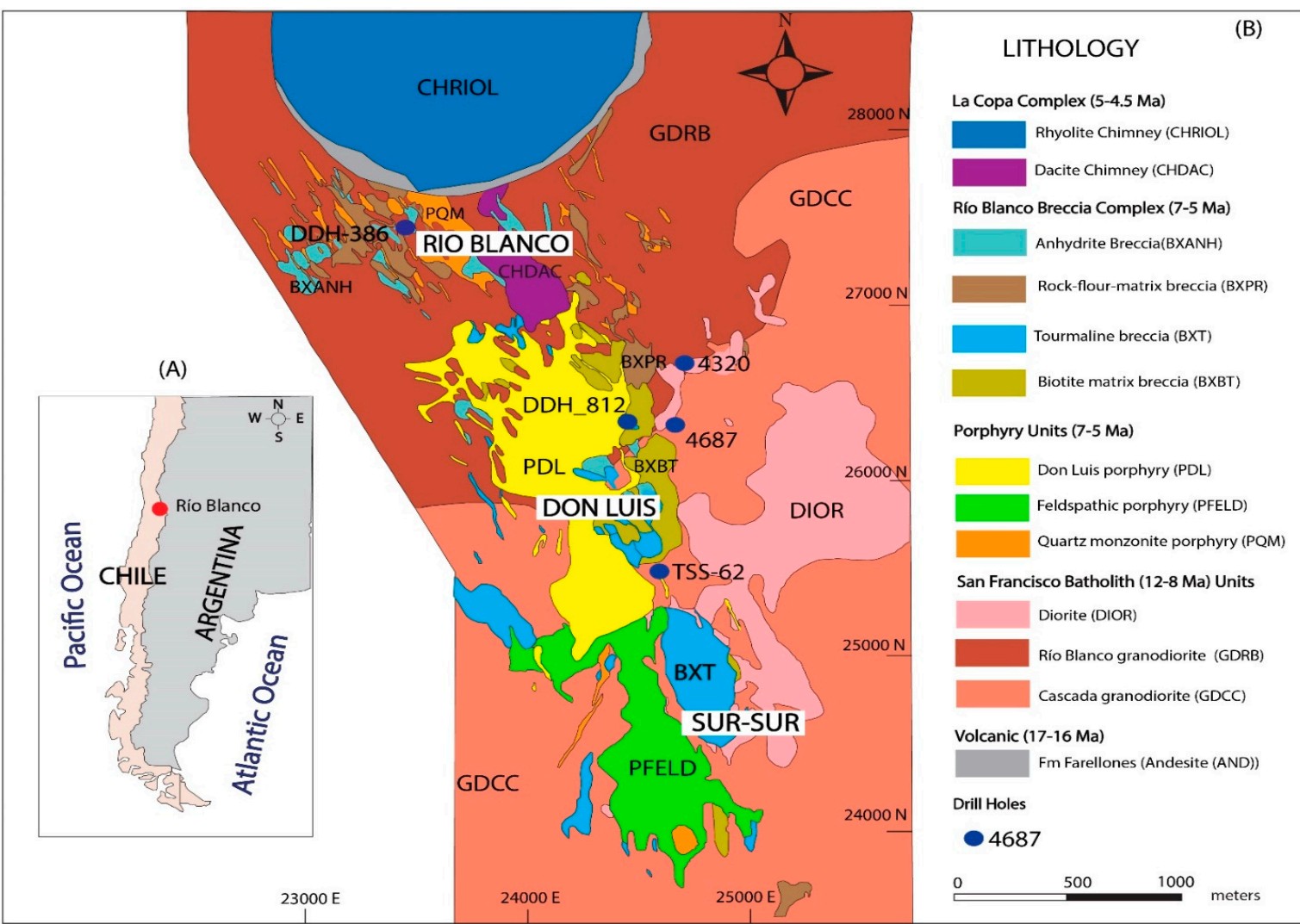

**Figure 1.** (**A**) Location of the Río Blanco porphyry Cu–Mo deposit in Central Chile. (**B**) Geology of the Río Blanco PCD, modified after Ferraz and Cruz 2011 [29].

The oldest rocks in the Río Blanco PCD correspond to andesite lavas and stratified basaltic andesites of the Farellones Formation [30], with reported U–Pb zircon ages of 17.2 ± 0.05 Ma [31] (Figure 1B). The Farellones Formation is intruded by the San Francisco Batholith (SFB), which comprises granodiorites, monzodiorites, and tonalites [32,33]. The SFB comprises several magmatic pulses, including the Río Blanco granodiorite (GDRB) and the Cascada granodiorite (GDCC), with reported U–Pb zircon ages of 11.96 ± 0.04 Ma and 8.4 ± 0.23 Ma, respectively [31].

The SFB is cut by porphyries of quartz monzonitic (PQM) and feldspathic (PFELD) compositions, with reported U–Pb zircon ages of 6.32 ± 0.09 Ma and 5.84 ± 0.03 Ma, respectively [31] (Figure 1B). Several hydrothermal breccia types are recognized in Río Blanco: igneous, igneous/hydrothermal, hydrothermal (biotite and tourmaline bearing), and rock–flour breccias [34]. The hydrothermal breccias are closely related to the porphyry units and the bulk of the copper sulfide mineralization occurs as cement in these breccias. The Don Luis porphyry (PDL) crosscuts the mineralized breccia complex and corresponds to a subvertical intrusion of dacitic composition (U–Pb zircon age of 5.23 ± 0.07 Ma [31]). The youngest unit in the district corresponds to the La Copa Volcanic Complex (CVLC), which is located in the northern part of the deposit. The CVLC comprises crystalline tuffs (CHDAC, Dacitic chimney; U–Pb zircon age of 4.57 ± 0.08 Ma) and lithic tuffs (CHRIOL, Rhyolitic chimney; U–Pb zircon age of 4.31 ± 0.05 Ma) [34]. Magmatic and hydrothermal fluid flow was channeled and focused by both sets of preexisting oblique structures (NE and NW-striking faults) and, in turn, fault rupture was driven by high fluid pressures [35].

Hydrothermal alteration at the Río Blanco PCD is characterized by a propylitic (chlorite–epidote) alteration that grades to biotite–chlorite and biotite towards the innermost part of the deposit. Potassic alteration is related to the porphyry phases, and it is characterized by a high content of biotite + potassium feldspar ± albite, and lower contents of anhydrite. Potassic alteration occurs at depth, with chalcopyrite ± bornite ± molybdenite and minor pyrite. This early alteration is identified by three types of veinlets: (i) Early Biotite Transitional (EBT) veins characterized by quartz + chalcopyrite ± bornite ± K feldspar ± anhydrite with biotite haloes; (ii) sinuous, A-type veinlets with quartz ± K feldspar ± chalcopyrite; and (iii) thicker, B-type veinlets with quartz ± molybdenite. The phyllic stage, overimposed on the potassic alteration event, is defined by the development of C-type gray–green sericite (GGS) veinlets with quartz + chalcopyrite, and a halo with phengite–celadonite ± K feldspar and abundant chalcopyrite ± bornite ± pyrite. This GGS alteration varies from strong to moderate to weak from the center to the outer zones of the deposit. In the shallowest parts of the deposit, a late-stage alteration event is recognized by the presence of D-type quartz–sericite veinlets (QS), composed of quartz + pyrite ± chalcopyrite and sericite ± clays (illite–kaolinite) halos with pyrite > chalcopyrite [36]. Late E-type veinlets are composed of quartz + carbonates (siderite–ankerite) + pyrite ± gypsum ± sphalerite ± tennantite ± enargite–luzonite ± galena ± bornite, with a sericite + clays halo.

## 3. Samples and Methods

CODELCO-Andina has a multi-element (51 elements) inductively coupled plasma mass spectrometry (ICP-MS) geochemical database of 10,140 samples (whole rock) from drill holes. A total of 28 core-samples from six drill holes that cross-cut the main mineralization-alteration zones at the Río Blanco deposit were collected for this exploratory study of sulfides. The samples were selected based on their Ag, Bi, and Te contents, with a focus on high grade material (>3 ppm Ag, 1.32 ppm Bi, and 0.24 ppm Te). The sulfide minerals studied here included chalcopyrite and bornite from the potassic alteration zone. Polished thick sections were inspected for the presence of micro- to nanometer-sized inclusions using a combination of conventional and field-emission scanning electron microscopy techniques. The SEM observations were carried out at the Andean Geothermal Center of Excellence (CEGA), Universidad de Chile, Santiago, Chile, using a FEI Quanta 250 SEM (Thermo Fisher Scientific) equipped with a secondary electron (SE) and backscattered electron (BSE) detectors, and an energy-dispersive X-ray spectrometer (EDS). The analytical parameters were: accelerating voltage of

15–20 kV and an emission current of ~80 μA, takeoff angle ~35°, spot beam was 4–5 μm in diameter, and a working distance of ~10 mm. Semi-quantitative EDS analyses were used to identify major elements in individual mineral phases. High-resolution imaging of micro- to nano-sized inclusions was achieved using field-emission scanning electron microscopy (FESEM). Observations were performed using a FEI Quanta 250 FEG at the Center for Research in Nanotechnology and Advanced Materials (CIEN) at the Pontificia Universidad Católica de Chile, Santiago, Chile. The FESEM is equipped with in-olumn detector (ICD) for SE and BSE, and an EDS detector. Operating conditions included an accelerating voltage of 20 kV, spot beam was ~4 μm in diameter, takeoff angle ~35°–37°, and the live time was 45 s and a working distance of ~10 mm.

In addition, major and minor element contents in chalcopyrite and bornite, and one palladium telluride inclusion were determined by EMPA using a JEOL JXA-8230 Superprobe at the LAMARX Laboratory of the Universidad Nacional de Córdoba, Argentina. Elements and X-ray lines used for the analysis were Hg (M$\alpha$), Te (L$\alpha$), Se (L$\alpha$), Bi (M$\alpha$), Au (M$\alpha$), S (K$\alpha$), Fe (K$\alpha$), Co (K$\alpha$), Zn (K$\alpha$), As (K$\alpha$), Ag (L$\alpha$), Pb (M$\beta$), Sb (L$\alpha$), Cu (K$\alpha$), Ni (K$\alpha$), Mn (K$\alpha$), Pt (M$\alpha$), Re (M$\alpha$), Pd (L$\alpha$), Rh (L$\alpha$), Ru (L$\alpha$), Ir (M$\alpha$), Os (M$\alpha$). Operating conditions included an accelerating voltage of 20 kV and a beam current of 20 nA, and the electron beam was ~1 μm in diameter. The counting time was 40 s for Hg, Te, Se, Bi, Au, S, Fe, Co, Zn, As, Ag, Pb, Sb, Cu, Ni, and Mn. The counting time was 20 s for Pt, Re, Pd, Rh, Ru, Ir, and Os. Standard specimens used for calibration were HgTe (for Hg and Te), NiSe (for Ni and Se), $Bi_2S_3$ (for Bi), $Au^0$ (for Au), $CuFeS_2$ (for Cu, Fe and S), $CoAs_3$ (for Co), ZnS (for Zn), NiAs (for As), $Ag^0$ (for Ag), PbS (for Pb), $Sb_2S_3$ (for Sb), $Mn^0$ (for Mn), $Pt^0$ (for Pt), $Re^0$ (for Re), $Pd^0$ (for Pd), $Rh^0$ (for Rh), $Ru^0$ (for Ru), $Ir^0$ (for Ir), and $Os^0$ (for Os).

## 4. Results

Representative EMPA analyses in chalcopyrite and bornite from the potassic alteration zone are reported in Table 1. Contents of Cu, Fe, and S in chalcopyrite ranged between 35.17 to 35.69 wt %, 29.10 to 29.88 wt %, and 34.09 to 34.46 wt % respectively. EMPA-WDS analyses showed that chalcopyrite contained Ag < 0.04 wt %, Au < 0.08 wt %, Co < 0.06 wt %, Ni < 0.01 wt %, Bi < 0.07 wt %, Hg < 0.12 wt %, Te < 0.03 wt %, Se < 0.03 wt %, As < 0.06 wt %, and Pb < 0.07 wt %. EMPA-WDS analyses of bornite ranged from 62.47 to 63.90 wt % Cu, 10.99 to 11.62 wt % Fe, and 25.48 to 26.02 wt % S. Other elements detected in bornite included Au < 0.05 wt %, Ag < 0.14wt %, Bi < 0.07 wt %, Hg < 0.10 wt %, Te < 0.03 wt %, Se < 0.02 wt %, As < 0.13 wt %, Pb < 0.07 wt %, Co < 0.02 wt %, and Ni < 0.01 wt %.

**Table 1.** Summary of representative electron microprobe analyses (EMPA) of chalcopyrite and bornite from the Rio Blanco deposit in (wt %). mdl = Minimum detection limits in wt %. b.d = below detection.

| Element | Cu | Fe | S | Au | Ag | Bi | Hg | Te | Se | Zn | As | Pb | Sb | Co | Ni | Total |
|---|---|---|---|---|---|---|---|---|---|---|---|---|---|---|---|---|
| mdl | 0.01 | 0.01 | 0.01 | 0.02 | 0.01 | 0.03 | 0.03 | 0.01 | 0.01 | 0.01 | 0.02 | 0.02 | 0.01 | 0.02 | 0.01 | |
| Sample DDH386-564 | | | | | | | | | | | | | | | | |
| C1-Cpy2 | 35.32 | 29.10 | 34.11 | b.d | b.d | b.d | 0.10 | 0.03 | b.d | b.d | 0.02 | 0.05 | b.d | b.d | b.d | 98.78 |
| C1-Cpy3 | 35.17 | 29.42 | 34.09 | b.d | b.d | b.d | b.d | 0.03 | b.d | b.d | b.d | 0.03 | b.d | 0.04 | b.d | 98.78 |
| C4-Cpy10 | 35.24 | 29.43 | 34.33 | b.d | b.d | 0.06 | b.d | b.d | b.d | b.d | b.d | b.d | b.d | b.d | b.d | 99.07 |
| C4-Cpy11 | 35.31 | 29.32 | 34.26 | 0.08 | b.d | b.d | b.d | b.d | 0.02 | b.d | b.d | b.d | b.d | 0.04 | 0.01 | 99.05 |
| C4-Cpy12 | 35.38 | 29.42 | 34.46 | b.d | b.d | b.d | 0.12 | b.d | 0.02 | b.d | b.d | 0.07 | b.d | b.d | b.d | 99.49 |
| C3-Cpy13 | 35.55 | 29.88 | 34.26 | b.d | 0.04 | 0.07 | 0.03 | 0.02 | 0.03 | b.d | 0.03 | 0.03 | b.d | 0.06 | b.d | 100.00 |
| C3-Cpy14 | 35.69 | 29.84 | 34.38 | b.d | b.d | 0.05 | 0.09 | b.d | 0.02 | b.d | 0.06 | 0.04 | b.d | 0.02 | b.d | 100.19 |
| C4-Bn8 | 63.85 | 10.99 | 25.55 | b.d | 0.14 | b.d | 0.10 | b.d | b.d | b.d | b.d | b.d | b.d | 0.02 | b.d | 100.65 |
| C4-Bn9 | 63.90 | 11.09 | 25.48 | b.d | 0.13 | b.d | 0.05 | 0.03 | b.d | b.d | 0.04 | b.d | b.d | b.d | b.d | 100.74 |
| C3-Bn15 | 62.67 | 11.62 | 25.76 | b.d | 0.04 | 0.06 | b.d | b.d | b.d | b.d | 0.08 | 0.07 | b.d | b.d | b.d | 100.30 |
| C3-Bn16 | 62.88 | 11.27 | 25.93 | b.d | 0.06 | b.d | 0.09 | b.d | 0.02 | b.d | 0.13 | b.d | b.d | 0.02 | 0.01 | 100.42 |
| C3-Bn17 | 62.47 | 11.28 | 26.02 | 0.05 | 0.06 | 0.07 | 0.10 | b.d | b.d | b.d | 0.12 | b.d | b.d | 0.02 | b.d | 100.22 |

| Element | Cu | Fe | S | Au | Ag | Bi | Te | Pd | Pt | Re | As | Rh | Ru | Ir | Os | Ni | Total |
|---|---|---|---|---|---|---|---|---|---|---|---|---|---|---|---|---|---|
| mdl | 0.01 | 0.01 | 0.01 | 0.02 | 0.01 | 0.06 | 0.03 | 0.03 | 0.04 | 0.03 | 0.02 | 0.02 | 0.02 | 0.03 | 0.03 | 0.02 | |
| Sample DDH386-564 | | | | | | | | | | | | | | | | | |
| C4-PdTe8 | 2.75 | 1.83 | 1.56 | 0.13 | b.d | 0.35 | 65.34 | 23.69 | 1.32 | b.d | b.d | b.d | b.d | b.d | 0.12 | b.d | 97.11 |

Detailed FESEM inspection of Ag-bearing chalcopyrite and bornite grains revealed the presence of micrometer to nanometer-sized mineral inclusions (Figures 2 and 3).

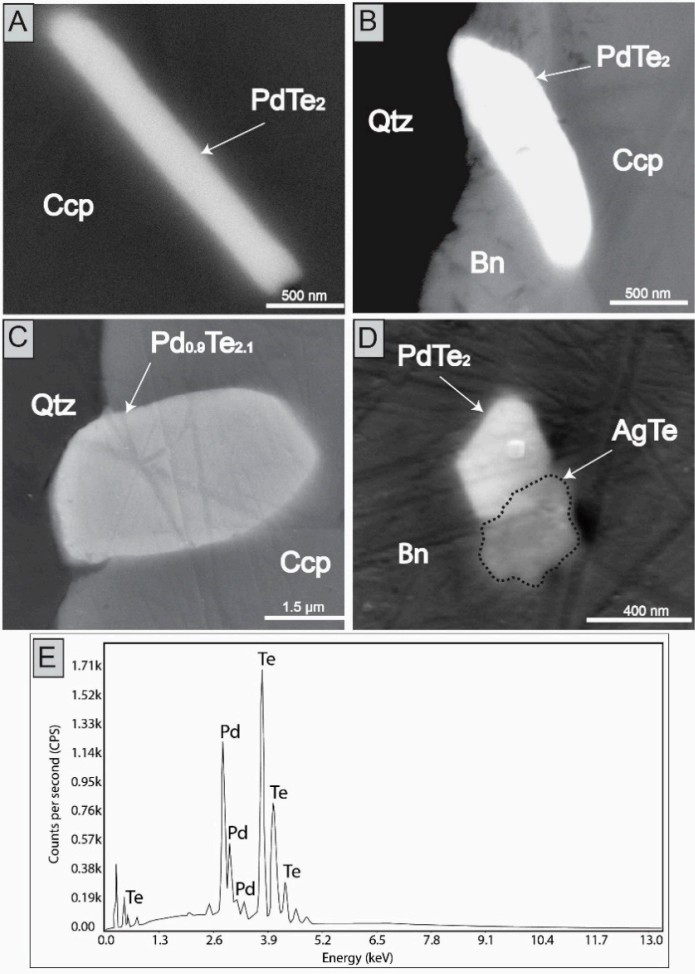

**Figure 2.** Field-emission scanning electron microscopy (FESEM) images of platinum group minerals (PGM) in chalcopyrite and bornite from Río Blanco. Palladium tellurides are shown in images (**A**–**D**). Images (**A**,**B**) were taken using the back-scattered electron detector (BSE), while images (**C**,**D**) were taken using the secondary electron (SE) detector. (**E**) shows the energy-dispersive X-ray spectrometer (EDS) spectrum of the Pd telluride grain in image (**C**). Ccp: chalcopyrite, Bn: bornite, Qtz: quartz, AgTe: silver telluride, $PdTe_2$: merenskyite.

The inclusions occur usually along grain boundaries (Figure 2B) and their morphologies varied from subhedral to tabular, with sizes ranging from ~400 nm to ~4 μm. Due to the sub-micrometer size of most inclusions, chemical characterization was carried out semi-quantitatively by means of FESEM-EDS. In most grains, Pd and Te were detected along with Cu, Fe, and S from the sulfide host matrix (Figure 2A,B,D). One quantitative EMPA-WDS spot analysis of a micrometer-sized inclusion (Figure 2C) showed ~23.7 wt % Pd and 65.3 wt % Te (Table 1). Despite the low total (~97 wt %), the stoichiometry is consistent with the mineral merenskyite ((Pd,Pt) (Bi,Te)$_2$). EMPA-WDS analysis of the same grain (Figure 2C) indicated the presence of Pt (1.3 wt %), Au (0.1 wt %), Os (0.1 wt %) and Bi (0.4 wt %). FESEM-EDS analyses of the other Pd–Te-bearing grains shown in Figure 2 were broadly consistent with merenskyite.

Mineral inclusions containing Au, Ag, Hg, and Te are shown in Figure 3. Most of these inclusions had anhedral to subhedral forms, some with rectangular and tabular/elongated shapes, and they occurred along grain boundaries (Figure 3A). Semi-quantitative FESEM-EDS analysis indicated that

the inclusions were most likely electrum (Au, Ag $\pm$ Hg) (Figure 3A–D), and Au–Ag–Te minerals such as petzite ($Au_3AgTe_2$) (Figure 3B).

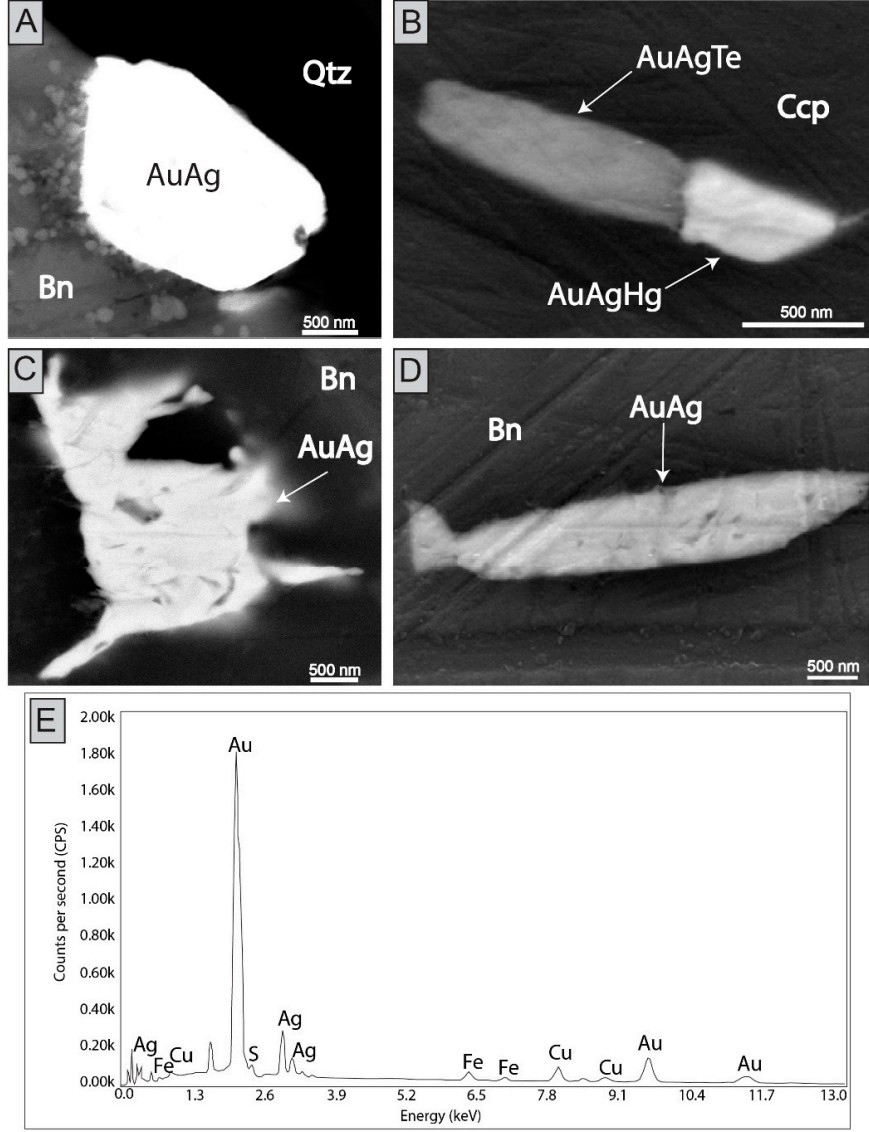

**Figure 3.** Field-emission scanning electron microscopy (FESEM) images of silver, gold, mercury, and tellurium mineral inclusions in chalcopyrite and bornite from Río Blanco. Electrum (Au,Ag) grains are shown images (**A–D**). Au–Ag–Te and Au–Ag–Hg phases are shown in (**B**). Images (**A,C**) were taken using the back-scattered electron (BSE) detector, while images (**B,D**) were taken using a secondary electron (SE) detector. (**E**) shows the EDS spectrum of the electrum grain shown in (**A**). Ccp: chalcopyrite, Bn: bornite, Qtz: quartz, AuAg: electrum, AuAgTe: possibly sylvanite or petzite.

## 5. Discussion

Anomalous PGE contents were previously reported in Cu–Fe sulfides and flotation concentrates from several porphyry Cu–Au deposits [4], where high Pd contents (130–1900 ppb) are associated with high Au contents (1–28 ppm). Additionally, Pašava et al., 2010 [9] showed in the Kalmarkyr porphyry Cu–Au–Mo deposit in Uzbekistan average concentrations of 55 ppb Pd, 5.5 ppb Pt, and 4.1 ppm Au for disseminated and stockwork—type high-grade Cu–Au–Mo mineralization. Economou-Eliopoulos et al., 2000 [5] reported relatively high Pd contents in the Skouries PCD in Greece, ranging between 60 and 200 ppb (average 110 ppb). Furthermore, Economou-Eliopoulos 2005 [8] concluded that PGE-bearing porphyries have similar characteristics,

including: (a) their association with alkaline or K-rich calc–alkaline systems; (b) the dominant occurrence of Pd-bearing minerals (merenskyite) within Cu sulfides, in association with Au–Ag tellurides; and (c) the association of Pd, Pt, and Au with magnetite–bornite–chalcopyrite assemblages, within the pervasive potassic alteration zones in the central parts of the deposits.

The features described above are broadly similar to the occurrences of PGM in the Río Blanco PCD, reported in this technical note. Our observations indicate that Pd, Pt, Au, Ag, and Te form micrometer to nanometer-sized mineral inclusions within chalcopyrite and bornite (Figures 2 and 3). PGM, Au, and Ag were found in early EBT veins characterized by quartz + chalcopyrite ± bornite ± K feldspar ± anhydrite with biotite haloes, and in type-A veinlets with quartz ± K feldspar ± chalcopyrite, which are characteristic veins of the penetrative potassium feldspar alteration zone. Gold is found as electrum (Au,Ag), whereas the most common silver-bearing mineral is hessite, which was observed intergrown with merenskyite ($PdTe_2$–$Ag_2Te$). Importantly, Tarkian et al., 1999 [4] analyzed mineral concentrates from the major Chilean porphyry Cu–Mo deposits (Río Blanco, Chuquicamata, Escondida, El Salvador and El Teniente). Among these, Río Blanco was the only deposit in which the authors did not report detectable concentrations of Pd and Pt. Since the aforementioned study focused on flotation concentrates, and no information was provided about the provenance of the samples within the deposit, it is likely that the apparent lack of Pd and Pt at Río Blanco was related to the fact that at the time, CODELCO was not mining the (deeper) potassic alteration zone, and the studied concentrate came from the upper portions of the deposit.

The mineralogical occurrence of Pd-bearing minerals (e.g., merenskyite, $((Pd,Pt)(Bi,Te)_2)$ at Río Blanco indicates that Pd was most likely introduced during the early potassic alteration stage from a high-temperature hydrothermal fluid. Xiong et al., 2000 [37] conducted a series of experiments on the solubility of Pd under hydrothermal conditions. High temperature, oxidized, and highly saline fluid conditions are thought to favor the hydrothermal transport of PGE. Xiong et al., 2000 [37] concluded that in the earlier stages of porphyry Cu–Mo formation, fluids are fully capable of transporting at least 10 ppb Pd. Recent experiments by Sullivan et al., 2018 [38] showed that Pd solubility increases with $fO_2$, indicating that Pd is dissolved in the silicate melt at nickel + nickel oxide buffer (NNO) between 0 and +1. Although the addition of Cl has a negligible effect on the solubility of Pd, experimental studies have shown that $PdCl_4^{-2}$ predominates between 25 and 300 °C. This complex is the main species at the $Cl^-$ concentrations observed in most hydrothermal fluids [39–41]. Since chloride ligands are generally invoked for Cu and Au transport in the porphyry environment [42,43], the close association of Cu-sulfides and noble metal (Au–Pd–Ag) inclusions at Río Blanco may be explained, in part, by a similar transport and deposition mechanism as their host minerals. An alternative mechanism of PGE enrichment may involve Pd–Bi–Te phases, which may remain molten above 489 °C [44]. The aforementioned authors provide evidence of high Pd and Bi concentrations in brine inclusions from the Skouries PCD in Greece, suggesting that Bi–Te melts may act as a collectors for PGE in high temperature hydrothermal fluids.

## 6. Concluding Remarks

The occurrence of Pd, Pt, and Au-bearing minerals in copper sulfides at Río Blanco opens new avenues of research aimed at assessing the noble metal content of the deposit. Also, the presence of PGE-bearing minerals in the potassic alteration zone at Río Blanco PCD poses questions that are relevant to the understanding of the speciation and solubility of noble metals during mineralization and hydrothermal alteration. An equally important question is the quantity and intensity of the mineralizing (hydrothermal) events that contributed to the deposition of the noble metals in PCDs. These questions are closely linked because the deposition of precious metals during different hydrothermal events could result in different incorporation forms into sulfides, i.e., solid solution vs. micro or nano-inclusions. Therefore, further micro-analytical studies are needed to address these questions, and to evaluate the sulfide phases formed under different physical and chemical conditions as hosts of PGE and other critical metals in porphyry Cu–Mo deposits.

**Author Contributions:** J.C. collected the samples and studied the sulfides using SEM, FESEM, and EMPA. All authors (J.C., M.R., F.B., J.J.V. and C.M.) discussed the results and evaluated the data. J.C., M.R. and F.B. wrote and organized the paper. J.J.V and C.M. provided geological information and granted access to the mine and drillcore libraries.

**Funding:** The work was supported by the Iniciativa Científica Milenio, which provided funding through grant #NC 130065 "Millennium Nucleus for Metal Tracing Along Subduction". Additional support was provided by the Andean Geothermal Center of Excellence (CEGA), FONDAP project #15090013. The Center for Research in Nanotechnology and Advanced Materials (CIEN) of the Pontifical Catholic University of Chile provided access to the FESEM used in this study, which was funded by FONDEQUIP Project EQM150101.

**Acknowledgments:** We thank the CODELCO Tech for providing a pre-doctoral Ph.D. scholarship to J. Crespo. The LAMARX Laboratory of the Universidad Nacional de Córdoba, Argentina, is acknowledged for granting access to the electron microprobe facility. This manuscript is dedicated to Romina, Naomi, and Tania.

**Conflicts of Interest:** The author declare no conflict of interest.

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
