# Peer review of "Critical Metal Particles in Copper Sulfides from the Supergiant Río Blanco Porphyry Cu–Mo Deposit, Chile"

_minerals, doi:10.3390/min8110519_

Round 1

Reviewer 1 Report

This is an excellent work pointing out another critical direction that the economic geology research on porphyry systems should move to in the future. I really look forward to seeing more detailed and continuing work on this topic from the authors in the future. Moreover, the manuscript is well organized, and the language is good. I therefore suggest that this manuscript is good for publication, and only a minor concern that the grade of the relavant metals of this deposit should be well addressed in the manuscript.

Author Response

Response to Reviewer 1 Comments

Point 1: This is an excellent work pointing out another critical direction that the economic geology research on porphyry systems should move to in the future. I really look forward to seeing more detailed and continuing work on this topic from the authors in the future. Moreover, the manuscript is well organized, and the language is good. I therefore suggest that this manuscript is good for publication, and only a minor concern that the grade of the relevant metals of this deposit should be well addressed in the manuscript. 

Response 1: We acknowledge the positive assessment of our manuscript. In fact, we are preparing a comprehensive manuscript that includes extensive EMPA and LA-ICP-MS data of the sulfides. Concerning grades, Rio Blanco hosts 0.78% and 0.021% Cu and Mo, respectively.

Reviewer 2 Report

Dear Sir,

I congratulate you on well-written article. It can be published in its current form. I have only one comment. I suggest (in the title) to replace the term "particles" with the term "mineral inclusions"

With the best regards

Author Response

Response to Reviewer 2 Comments

Point 1: I congratulate you on well-written article. It can be published in its current form. I have only one comment. I suggest (in the title) to replace the term "particles" with the term "mineral inclusions"

Response 1: We acknowledge the positive feedback to our manuscript. However, we believe that the current title adequately reflects what we want to express in this technical note.

Reviewer 3 Report

Review of Minerals, Manuscript Number 379918.

Title: Critical Metal Particles in Copper Sulfides from the Supergiant Río Blanco Porphyry Cu-Mo Deposit, Chile.

Author: Jorge Crespo et al.

This paper presents a short and interesting mineralogical study of platinum-group minerals and related phases in Fe-Cu sulfides from the Río Blanco porphyry deposit. The paper is well written and is concise. Briefly, the authors present new data about the identification of nano- and micro particles of PGM and Ag-Au-bearing phases within primary sulfides. They discuss these results in terms of mobility of PGE during altering/mineralizing processes in porphyry systems. In my opinion, this paper will result of interest for readers of Minerals, so I recommend its publication. I have only done a few constructive suggestions to this good manuscript.

Minor comments

- General comment: what about pyrite as carrier of PGE/Au? Several studies in magmatic sulfide systems have shown that pyrite is much more important as host of PGE (particularly Pt-Rh) and Au than chalcopyrite. They often exhibit beautiful zoning in PGE from core to rim in close relationship with As and Sb variations. I think that it would be very interesting to check pyrite.

Abstract

- Line 21: “… copper sulfides from one of the largest PCD in the world”.

- Line 23-24: as expressed it seems that inclusions are pure noble metals when they are really phases bearing Te; I mean, they are tellurides. The authors say that they investigate the mineralogical expression of Pt (line 20), but they do not report any Pt-bearing phase (lines 24 …).

- Line 29. Maybe, too much optimistic. I have serious doubts that the sulfides in these deposits could become a viable source of these metals.

Introduction

- Line 37. Use symbol for elements instead of word? Mo, Re, …

- Lines 44-45: gangue sulfides? What do you mean?

- Line 64: alkaline instead of alkalic

- Lines 66-67. That is!! Pyrite is much more interesting tan Fe-Cu sulfides.

- Lines 106-122: the authors describe the presence of chalcopyrite, pyrite and bornite in different hydrothermal alteration zones at the Río Blanco PCD (for example, potassic, phyllic, late-stage, …), but they only analyze chalcopyrite and bornite from the potassic alteration zone. Why? Maybe, it is because this is a preliminary study. Ok, but the reader will have this doubt and maybe the authors should explain the current point of the research.

Samples and methods

- Lines 127-128. For sampling, whole rock concentrations of Ag, Bi and Te are used as indicative of high PGE contents, right? Is there no bulk concentration data for PGE, particularly Pt and Pd?

Results

- Lines 158-159: Caution; there are only two analyses with Au contents higher than detection limit. Here, there is maybe a problem with redaction. Ok, some chalcopyrite grains contain Au, but since most chalcopyrite grains have Au below detection limit (we do not know exactly if these grains contain or do not contain gold), I think that it is not totally correct write “… chalcopyrite… contain significant amounts of Au (< 800 ppm)”. Just the same for the rest of elements. On the other hand, electron microprobe is not the best technique to study the trace element abundances. Have the authors considered using LA-ICP-MS?

- General comment: the abstract says that the occurrence of Pt is reported but there is no information about the presence of Pt as PGM.

Discussion

- Line 201. Add et al after Eliopoulus.

- Line 208. There is no information about Pt, so Pt should be remove out of results.

- Line 223. “… Pd was most likely introduced during the early potassic alteration stage”. From where?

Figures.

- Figure 1: the geological map is too small in comparison with the map with the geographic location. My recommendation would be increasing the size of the geological map. The legend of the map should be consistent with the description in the text. I mean, in the text I read Fm. Farellones and San Francisco batholith, but it is not easy to recognize these names in the map because the legend does not include them.

Table

- Table 1. As I have told above for lines 158-159, it is a bit confusing to say that “chalcopyrite has significant amounts of Ag (<1400 ppm)” when, in fact, the table 1 shows that only one chalcopyrite grain analyzed contained a quantity of silver above the detection limit of the technique. Similar observation for Ni (0.01 wt. %; that is 100 ppm, which is the value of its limit of detection?)

October 2018

Author Response

Response to Reviewer 3 Comments

Point 1: Minor comments

 - General comment: what about pyrite as carrier of PGE/Au? Several studies in magmatic sulfide systems have shown that pyrite is much more important as host of PGE (particularly Pt-Rh) and Au than chalcopyrite. They often exhibit beautiful zoning in PGE from core to rim in close relationship with As and Sb variations. I think that it would be very interesting to check pyrite.

Response 1: In lines 59-65, we cite the work of Hanley et al., 2009, which documented the occurrence of Pd and Pt within pyrite. Based on this, we initially targeted pyrite at Río Blanco but so far we only found PGE/Au mineral particles within chalcopyrite and bornite from the potassic alteration zone. We are currently focusing on measuring PGE/Au concentrations in pyrite using LA-ICP-MS. 

Point 2: Line 21: “… copper sulfides from one of the largest PCD in the world”.

Response 2: Fixed (now line 20)

Point 3: Line 23-24: as expressed it seems that inclusions are pure noble metals when they are really phases bearing Te; I mean, they are tellurides. The authors say that they investigate the mineralogical expression of Pt (line 20), but they do not report any Pt-bearing phase (lines 24 …).

Response 3: In lines 23-24, we mention the occurrence of PGM inclusions, which are mostly tellurides, such as merenskyite [(Pd, Pt) (Bi, Te)2], a mineral that contains Pt in its chemical formula. Table 1 shows one quantitative EMPA-WDS spot analysis of a micrometer-sized inclusion of merenskyite with 1.32 wt% Pt.

Point 4: Line 29. Maybe, too much optimistic. I have serious doubts that the sulfides in these deposits could become a viable source of these metals.

Response 4: Despite the relatively low Pd and Pt contents, the large volume of exploited copper sulfides would make Río Blanco attractive to explore as a possible source of noble metals (now line 28). But we get the point, and we have toned down the last sentence of the abstract.

Point 5: - Line 37. Use symbol for elements instead of word? Mo, Re, …

Response 5: Fixed (now line 34)

Point 6: - Lines 44-45: gangue sulfides? What do you mean?

Response 6: Fixed (now line 42)

Point 7: - Line 64: alkaline instead of alkalic

Response 7: Fixed (now line 60)

Point 8: - Lines 66-67. That is!! Pyrite is much more interesting than Fe-Cu sulfides.

Response 8: We agree that pyrite has the potential to host noble metals such as Au and PGE at Río Blanco. We are currently exploring this possibility using LA-ICP-MS methods.

Point 9: - Lines 106-122: the authors describe the presence of chalcopyrite, pyrite and bornite in different hydrothermal alteration zones at the Río Blanco PCD (for example, potassic, phyllic, late-stage, …), but they only analyze chalcopyrite and bornite from the potassic alteration zone. Why? Maybe, it is because this is a preliminary study. Ok, but the reader will have this doubt and maybe the authors should explain the current point of the research.

Response 9: This study is part of a larger project that aims to constrain the occurrence of Ag and associated critical metals in the Río Blanco PCD. Hence, our technical note aims to report our exploratory FESEM study of the copper sulphides at Río Blanco, with emphasis on the potassic zone. We focused on chalcopyrite and bornite because these are the two main phases that carry Ag and associated metals within the deposit, and the highest grades are found in the potassic zone.

Point 10: Samples and methods

 - Lines 127-128. For sampling, whole rock concentrations of Ag, Bi and Te are used as indicative of high PGE contents, right? Is there no bulk concentration data for PGE, particularly Pt and Pd?

Response 10: Correct, we used these elements as a proxy for PGEs since Codelco’s ICP-MS database did not include PGEs. But we are doing it…  

Point 11: Results

 - Lines 158-159: Caution; there are only two analyses with Au contents higher than detection limit. Here, there is maybe a problem with redaction. Ok, some chalcopyrite grains contain Au, but since most chalcopyrite grains have Au below detection limit (we do not know exactly if these grains contain or do not contain gold), I think that it is not totally correct write “… chalcopyrite… contain significant amounts of Au (< 800 ppm)”. Just the same for the rest of elements. On the other hand, electron microprobe is not the best technique to study the trace element abundances. Have the authors considered using LA-ICP-MS?

- General comment: the abstract says that the occurrence of Pt is reported but there is no information about the presence of Pt as PGM.

Response 11: Accepted (now lines 149-156)

Point 12: - Line 201. Add et al after Eliopoulus.

Response 12: This work was done by only one author, hence et al. does not correspond. (now line 195)

Point 13: -Line 208. There is no information about Pt, so Pt should be remove out of results.

Response 13: Table 1 shows one quantitative EMPA-WDS spot analysis of a micrometer-sized inclusion of merenskyite [(Pd, Pt) (Bi, Te)2], with 1.32 wt% Pt, 0.12 wt% Os.

Point 14: - Line 223. “… Pd was most likely introduced during the early potassic alteration stage”. From where?

Response 14: Fixed (now line 215-216) Pd was most likely introduced during the early potassic alteration stage from a high-temperature hydrothermal fluid.

Point 15: Figures.

 - Figure 1: the geological map is too small in comparison with the map with the geographic location. My recommendation would be increasing the size of the geological map. The legend of the map should be consistent with the description in the text. I mean, in the text I read Fm. Farellones and San Francisco batholith, but it is not easy to recognize these names in the map because the legend does not include them.

Response 15: We agree. The figure was modified accordingly.

Point 16: Table¨

 - Table 1. As I have told above for lines 158-159, it is a bit confusing to say that “chalcopyrite has significant amounts of Ag (<1400 ppm)” when, in fact, the table 1 shows that only one chalcopyrite grain analyzed contained a quantity of silver above the detection limit of the technique. Similar observation for Ni (0.01 wt. %; that is 100 ppm, which is the value of its limit of detection?)*

Response 16: Fixed (now lines 149-153)
